# The Preparation and Physiochemical Characterization of *Tenebrio molitor* Chitin Using Alcalase

**DOI:** 10.3390/molecules28073254

**Published:** 2023-04-05

**Authors:** Hyemi Kim, Hyeongyeong Kim, Yejin Ahn, Ki-Bae Hong, In-Woo Kim, Ra-Yeong Choi, Hyung Joo Suh, Sung Hee Han

**Affiliations:** 1Department of Integrated Biomedical and Life Sciences, Graduate School, Korea University, Seoul 02841, Republic of Korea; hm1004486@nate.com (H.K.); hyunkyung999@gmail.com (H.K.); ahnyj708@gmail.com (Y.A.); suh1960@korea.ac.kr (H.J.S.); 2Department of Food Science and Nutrition, Jeju National University, Jeju 63243, Republic of Korea; kbhong@jejunu.ac.kr; 3National Institute of Agriculture Science, Wanju 55365, Republic of Korea; kiw0601@korea.kr (I.-W.K.);; 4Transdisciplinary Major in Learning Health Systems, Department of Healthcare Sciences, Graduate School, Korea University, Seoul 02841, Republic of Korea; 5Institute of Human Behavior & Genetics, Korea University, Seoul 02841, Republic of Korea

**Keywords:** chitin, chitosan, mealworms, *Tenebrio molitor*, Alcalase, biomedical products

## Abstract

Chitin is mostly produced from crustaceans, but it is difficult to supply raw materials due to marine pollution, and the commonly used chemical chitin extraction method is not environmentally friendly. Therefore, this study aims to establish a chitin extraction process using enzymes and to develop edible insect-derived chitin as an eco-friendly new material. The response surface methodology (RSM) was used to determine the optimal conditions for enzymatic hydrolysis. The optimal conditions for enzymatic hydrolysis by RSM were determined to be the substrate concentration (7.5%), enzyme concentration (80 μL/g), and reaction time (24 h). The solubility and DDA of the mealworm chitosan were 45% and 37%, respectively, and those of the commercial chitosan were 61% and 57%, respectively. In regard to the thermodynamic properties, the exothermic peak of mealworm chitin was similar to that of commercial chitin. In the FT-IR spectrum, a band was observed in mealworm chitin corresponding to the C=O of the NHCOCH_3_ group at 1645 cm^−1^, but this band showed low-intensity C=O in the mealworm chitosan due to deacetylation. Collectively, mealworm chitosan shows almost similar physical and chemical properties to commercial chitosan. Therefore, it is shown that an eco-friendly process can be introduced into chitosan production by using enzyme-extracted mealworms for chitin/chitosan production.

## 1. Introduction

Chitin, a polymer of *N*-acetyl-d-glucosamine, is the second most widely distributed polysaccharide in nature, right after cellulose. Chitin, which exists in nature, can be obtained from various biological resources, such as crustaceans in the sea, cell walls of fungi, and exoskeletons of insects. Among the various biological resources, chitin has been mainly produced from crustaceans. Chitin and chitosan have been mainly used as water treatment agents, such as dehydrating agents and coagulants in wastewater treatment [1]. It is known to improve vascular disease prevention and has anticancer effects that suppress cancer cell proliferation, as well as suppressing blood pressure rise, proliferating effective bacteria in the intestine, and activating cells by adsorbing and excreting excessive harmful cholesterol in the body. As the immune-enhancing activity was recently reported, it began to be applied to food as a health functional food material [2]. In fields other than food, it is used for various purposes, such as feed for livestock and fisheries, insecticides, disinfectants, sewage treatment agents, cosmetic materials, various films or wraps, medical artificial skin, and surgical sutures [3].

Food development is accelerating due to the registration of food ingredients made from edible insect, and it is possible to materialize food and medicine through high-tech convergence. In particular, edible insect cuticles are composed of chitin, lipid, and several compounds [4]. The chemical chitin extraction method using acids and bases is an extraction method generally used to date. Moreover, the chitin yield from edible insects is higher than that from crustaceans [5]. However, chemical extraction has several negative effects on the environment, and proteins cannot be recovered during the extraction process [6]. Chemical treatment methods generate HCl and NaOH wastewater used in extracting chitin or producing chitosan. In addition, salts and protein wastewater generated during the neutralization of HCl and NaOH are the main causes of environmental pollution [7,8]. Eco-friendly extraction methods have been applied to solve this problem. Among the eco-friendly extraction methods, enzymes can prevent irregular deacetylation and molecular weight reduction due to chemical treatment [9]. Because the process is carried out under mild conditions, energy and chemical reduction effects can be explored and maximized. In addition, it has the advantage of being able to recover and reuse proteins [10].

Insects continuously synthesize and decompose chitin, as they must periodically remodel their structures for growth and development during their life cycle [11]. Insects also have a short generation period and a short breeding period due to their good reproductive ability and small size, so they do not occupy a lot of breeding space, so labor for breeding can be preserved. In addition, environmental pollution for breeding can be reduced, and due to a decrease in the catch of snow crabs, which are major chitin and chitosan sources, due to sea pollution, it is urgent to find new chitin and chitosan sources [12]. Therefore, edible insects have infinite value as natural resources for chitin and chitosan. The mealworm (*Tenebrio molitor* L.), a representative edible insect, is a type of insect widely distributed in Korea and around the world; it belongs to the family Coleoptera [13]. Mealworms have a fast life cycle of three months on average [14]; therefore, mass production is possible in a short period. Mealworms are relatively easy to industrialize because they have a high reproductive rate and breeding conditions are not difficult [15]. During the metamorphosis of mealworm larvae into pupae, an exoskeleton is created, the main component of which is chitin [16,17]. However, there are few studies on the production of chitin from mealworm, an edible insect, as well as on eco-friendly and economical (the protein removed during the chitin extraction can be removed and recycled) methods using the enzyme.

In this study, Alcalase was selected among the commercial proteases, and the optimal conditions for protein degradation by Alcalase were established through the response surface methodology (RSM). In addition, the physicochemical properties of chitin and chitosan from which proteins were removed by Alcalase under optimal reaction conditions were evaluated.

## 2. Results and Discussions

### 2.1. Protease Screening for Protein Removal from Mealworms

When the mealworms were treated with proteolytic enzymes, the DP was calculated by measuring the amount of AN based on the reaction time. There was a tendency for the DP to increase due to an increase in the amount of AN as the reaction time increased (Figure 1). When the hydrolysis was performed for 24 h, the hydrolysate produced by Alcalase showed the highest AN content at 229.42 (mg/g of sample). Alcalase treatment for 24 h resulted in a high protein removal rate (63.16%). Lucas’ study also used Alcalase to remove proteins, and the DH at that time was reported to be 51.33% [5], and our results were higher than that, although the enzyme reaction conditions were different. When a protein hydrolysate containing active peptides was prepared from brown mealworm larvae, the yield of peptides of 3 kDa or less in the hydrolysate by Alcalase was 42.1%, which was higher than the hydrolysis by Flavourzyme [18]. In this study, Alcalase showed a higher AN content in the hydrolysate than Flavourzyme. In addition, it was reported that treatment with plant proteases, such as bromelain and papain, slowly progressed the hydrolysis of brown mealworm larvae. The Collupulin used in this study is an enzyme corresponding to papain, and unlike previous reports, the AN content recovered during the hydrolysis was relatively high. This difference seems to be due to the differences in the substrate specificity of the enzyme. In previous studies [18,19], it was confirmed that the protein hydrolysate using Alcalase contained a large amount of low-molecular-weight peptides and had a significantly higher degree of hydrolysis than other enzyme groups. Alcalase seems to contribute more to producing lower-molecular-weight peptides from insect proteins than other proteases. Therefore, Alcalase, with the highest DP, was selected as the optimal enzyme to remove mealworm proteins.

### 2.2. Establishment of Optimal Alcalase Hydrolysis Conditions for Mealworm Protein Removal by RSM

Through the RSM design, a regression equation for optimal chitin extraction conditions using enzymes was obtained. All the main effects, including linear and interaction effects, were calculated for the model. The regression coefficients were as follows:Y = −16.5038 − 0.8600X + 0.4812Y + 15.6862Z + 0.2247X^2^ − 0.0014Y^2^ − 0.4522Z^2^ − 0.0178XY + 0.0339YZ − 0.1470XZ (R^2^ = 0.957)

We obtained the predictive conditions of 7.5 g of the substrate concentration, 80 μL of the Alcalase concentration, and 24 h of time through the regression model. The amino-nitrogen content (mg/g of sample) under the predictive condition was 163.17 mg/g. As a result of the actual experiment, the highest amino-nitrogen content (sample 162.99 mg/g) was found under the same conditions as the predicted conditions. The optimal conditions for chitin and chitosan production were as follows: substrate concentration, 7.5 g; Alcalase concentration, 80 μL; and time, 24 h (Table 1). In the 3D graph, the substrate and enzyme concentrations show low amino-nitrogen content (blue), which is a dependent variable at low concentrations, and high content (red) at optimal concentrations, respectively. This showed the same trend for the enzyme concentration time and substrate concentration time (Figure 2). In constructing this model, a preliminary experiment was first conducted based on references to the optimal reaction conditions of the enzyme to set the range of the substrate concentration, enzyme concentration, and reaction time factors. Before this experiment, we set the range of each factor based on the results of preliminary experiments with a substrate concentration of 2% to 15%, an enzyme concentration of 10 to 120, and a reaction time of 3 to 48 h. Finally, for the design, the substrate concentration (5, 7.5, 10, and 12.5%), amount of enzyme added per substrate (25, 50, 80, and 100 μL/g), and reaction time (6, 8, 12, and 24 h) to determine the optimal hydrolysis conditions were set as the independent variables. Therefore, the conditions of the results of this study seem to be very optimal results. The lack of conformity was confirmed through the variance analysis of the RSM, and each value is as follows. With DF: 5, SeqSS: 863.5, AdjSS: 863.5, AdjMS: 172.70, F-value: 19.37, and *p*-value: 0.000, the optimal reaction conditions we obtain are considered meaningful.

The organically derived chitin and chitosan remain very relevant and significant in the biomedical world today. In this study, they are produced by optimizing the conditions for the enzyme concentration, substrate concentration, and specific durations for scientific methods (Table 1). In addition, the produced chitin and chitosan express the same physicochemical characteristics. The high-quality chitin and chitosan can serve various purposes in the biomedical field and beyond.

### 2.3. FT-IR Spectroscopy of Chitin and Chitosan

Figure 3 shows the measurement results of the infrared absorption spectra of the chitin and chitosan in the range 380 cm^−1^–4000 cm^−1^. The infrared absorption spectrum analysis of the chitin, commercial chitin (C-1) derived from shrimp shells, and chitin (C-2) derived from mealworm larvae show similar spectra. The absorption peak, characteristic of chitin, showed amide I at the 1650 (C-1)/1626 (C-2) cm^−1^ wavelength and amide II bands at the 1557 (C-1)/1516 (C-2) cm^−1^ wavelength. Strong peaks due to OH stretching, CH stretching, and amide III vibration were observed at 3287 (C-1)/3262 (C-2) cm^−1^, 2875 (C-1), 2853 (C-2) cm^−1^, and 1374 (C-1)/1379 (C-2) cm^−1^ [20,21]. The signal of the chitin prepared from the mealworm larvae was similar to that of the α-chitin derived from shrimp shells. Therefore, chitin prepared from mealworm larvae contains α-chitin. However, the peak showing a distinct difference from the shrimp shell-derived α-chitin was higher at 2878 cm^−1^, indicating C–H stretching. Chitin derived from insects has the typical crystal structure of α-chitin, regardless of the type of insect [22,23].

Figure 3 shows the FT-IR spectra of the chitosan (CS-1) derived from shrimp shells and chitosan (CS-2) derived from mealworm larvae. In CS-2, peaks due to C–H symmetry and asymmetric stretching were confirmed at wavelengths of 2918 and 2849 cm^−1^, respectively, but these peaks were not confirmed in the CS-1 chitosan. The small peaks at 1657 (CS-1)/1659 (CS-2) cm^−1^ (C=O stretch of amide I) and 1307 (CS-1)/1307 (CS-2) cm^−1^ (C–N stretch of amide III) indicated the presence of *N*-acetyl groups. A peak corresponding to the N–H bending of amide II and a peak corresponding to the N–H bending of the primary amine were detected at the 1552 (CS-1)/1553 (CS-2) cm^−1^ and 1621 (CS-1)/1620 (CS-2) cm^−1^ wavelengths, respectively [24]. CH2 bending and CH3 symmetrical deformations were confirmed by the presence of bands at 1420 (CS-1)/1421 (CS-2) cm^−1^ and 1375 (CS-1)/1375 (CS-2) cm^−1^, respectively. The absorption band at 1153 (CS-1)/1154 (CS-2) cm^−1^ can be attributed to the asymmetric stretching of the C–O–C bridge. The small peak at 1258 (CS-1)/1262 (CS-2) cm^−1^ is presumed to be due to the bending vibration of the hydroxyl group present in chitosan [25]. The peak at 894 (CS-1)/894 (CS-2) cm^−1^ corresponds to CH bending out of the monosaccharide ring plane. The bands at 1067 (CS-1)/1067 (CS-2) cm^−1^ and 1010 (CS-1)/1008 (CS-2) cm^−1^ correspond to C–O stretching. These peaks were also observed in the spectra of the other chitosan samples [25,26]. The chitosan derived from mealworm larvae (CS-2) showed an FT-IR spectrum similar to that of the shrimp shell-derived chitosan (CS-1). As such, the difference in the crystallinity according to the sample seems to be due to the difference in deacetylation [27]. Deacetylation affects crystalline and amorphous regions, directly affecting the strength of the crystal.

### 2.4. Thermodynamic Properties of Chitin and Chitosan Using DSC

To analyze the physical and thermodynamic properties of the chitin and chitosan samples derived from mealworms and the commercial chitin and chitosan samples, their thermodynamic properties were measured using DSC. In the exothermic reaction of the commercial chitin, the *T_max_*_1_ was 143.29 °C, and that of the enzyme-treated chitin was 153.76 °C. This is characterized by the removal of the residual moisture present in the sample and the change in the aggregation state, indicating the initiation of material combustion [28,29]. The energy (enthalpy, Δ*H*) for thermal decomposition is required to break the bonds that stabilize the chitin or chitosan structure. As for the endothermic peaks of the chitin and chitosan, sample degradation-related endothermic reactions were observed at 325–398 °C and 280–372 °C, respectively (Table 2). The enthalpy required for the endothermic reaction of chitosan is higher than that of chitin (Table 2). It has been reported that the difference in enthalpy between commercial chitin/chitosan and chitin/chitosan by enzymatic treatment is due to the difference in purity and the amount of NaOH used for the protein or deacetylation [29].

Even for the mealworm chitosan, the measured values for the endothermic and exothermic reactions were similar to those for the commercial chitosan (Table 2). The thermal stability and thermodynamic properties of the mealworm chitin, chitosan, commercial chitin, and chitosan are assumed to be similar.

### 2.5. Degree of Deacetylation (DDA) and Solubility of Chitosan

The DDA is an important indicator used to determine the chitosan type. The DDA of the chitosan derived from the mealworm was 37.34%, and that of the commercial chitosan was 57.58% (Table 3). In the chitosan derived from the larval exoskeleton, a relatively low DDA was reported, with a DDA of 34–72% [20]. The DDA of chitosan in insects is lower than that in crustaceans [30]. These measurements are related to the chitin content extracted from insects, which is thought to differ depending on the skeletal region or life cycle stage of the insect [5]. It is known that the arrangement of monomers and acetyl groups has a more important effect on the function of chitosan than the degree of simple deacetylation [31].

The solubility of the chitosan derived from the mealworm was 45.94%, and the solubility of the commercial chitosan was 61.34% (Table 3). The molecular structure of chitin is an intermolecular hydrogen-bond network that is insoluble in general organic and inorganic solvents, and deacetylated chitosan is dissolved in acidic solvents. In general, the solubility increases with a high degree of deacetylation but can vary depending on the cause [32]. The solubility of chitosan is affected by the pH, molecular weight, ionic strength, and temperature [33], but commercial chitosan with a high DDA value is highly soluble; therefore, solubility seems to have a close relationship with the DDA.

## 3. Materials and Methods

### 3.1. Materials

The Rural Development Administration provided vacuum-dried mealworms. The enzymes used for protein removal were Alcalase, Protamex, and Flavourzyme, purchased from Novozymes (Bagsvaerd, Denmark). Multifect PR7L and Collupulin MG were purchased from Biosion Biochem Co. (Sungnam, Republic of Korea) and DSM Food Specialties (Heerlen, The Netherlands), respectively. For other experiments, experiments were carried out using reagents of first class or higher.

### 3.2. Mealworm Protein Hydrolysis and Deacetylation

To produce chitin and chitosan derived from mealworm larvae, proteins were first removed using proteolytic enzymes. After suspending 5% of mealworms in phosphate buffer (50 mM, pH 7.5), 2% of the enzyme was added compared to mealworm powder, and the enzyme reaction was performed while shaking (130 rpm) at 50 °C. After the enzyme reaction was complete, it was heated at 95 °C for 15 min to inactivate the enzyme. Next, the heat-treated enzyme reaction solution was centrifuged (4000 rpm, 4 °C, 20 min) to separate the supernatant from the precipitate. The precipitate was then washed twice with DW and used as the chitin. A 10-fold amount of 40% NaOH solution was added to the protein-free chitin and reacted at 90 °C. After 8 h of reaction, it was washed with distilled water until neutral (pH 7.0) and then filtered (Whatman filter paper No. 1). The filtrate was dried at 60 °C for 12 h to obtain chitosan powder.

### 3.3. Determination of Degree of Deproteinization (DP)

The amino-nitrogen content of the mealworm treated with enzymatic digestion and the amino-nitrogen content of the acid hydrolysate of the mealworm was measured, and the DP was calculated as follows [34]:DP(%) = [(L*_t_* − L_0_)/(L_max_ − L_0_)] × 100

L*_t_*: amount of α-amino-nitrogen released at time *t.*

L_0_: amount of α-amino-nitrogen in the original sample.

L_max_: maximum amount of α-amino-nitrogen in the sample obtained after acid hydrolysis.

Acid hydrolysis of mealworms was performed at 100 °C for 24 h using 6 N HCl [34]. The acid hydrolysate was centrifuged (4000× *g*) to obtain the supernatant. The supernatant was neutralized with 6 N NaOH before measuring the amino-nitrogen content of the supernatant.

### 3.4. Optimization of Protein Removal Conditions by Alcalase by Response Surface Methodology (RSM)

To optimize the hydrolysis conditions of Alcalase, a protein removal enzyme of mealworms, the experiment was performed using an incomplete factorial design (central composite design) by using RSM. The substrate concentration (5, 7.5, 10, and 12.5%), amount of enzyme added per substrate (25, 50, 80, and 100 μL/g), and reaction time (6, 8, 12, and 24 h) to determine the optimal hydrolysis conditions were set as the independent variables (Table 4). The dependent variable was set to amino-nitrogen content (mg/g of sample), and hydrolysis was carried out randomly 40 times. Data analysis was performed using a response surface regression statistical analysis system (Minitab^®^ Software, version 14; Minitab Inc., State College, PA, USA). A second-order model was employed to fit the data, in which each coded factor was in the range of −168,176, 0, and +168,176. The second-order equation is as follows:Y= A + aX + bY + cZ + dX^2^ + eY^2^ + fZ^2^ + hYZ + iXZ
where X (concentration of substrate), Y (concentration of enzyme), and Z (time, h) are variables in the model, and this second-order model was employed to fit the data for Y.

### 3.5. α-Amino Nitrogen (AN) and Crude Protein Analysis

To measure the amount of protein removed, the α-amino nitrogen content of the supernatant of the hydrolysate treated with the enzyme was measured using the 2,4,6-trinitrobenzensulfonic acid (TNBS) method [35]. In this case, l-leucine was used as the standard.

### 3.6. Solubility (%) of Chitosan Derived from Mealworm

To measure the solubility of chitosan, 0.01 g of the sample was dissolved in 10 mL of 2% acetic acid solution by vortexing for 5 min, and the precipitate was recovered by centrifugation (4000 rpm, 10 min, 4 °C). After the recovered precipitate was dried in an oven (60 °C, overnight), the solubility of chitosan was calculated by measuring the weight before and after dissolution [5].

### 3.7. FT-IR Spectroscopy of Chitin and Chitosan

The FT-IR spectra of chitin and chitosan were measured at a wavelength of 4000–380 cm^−1^ by the UATR technique, using PerkinElmer^®^ Spectrum™100 (Waltham, MA, USA) [18]. This instrument has a mirror speed of 0.2 cm^−1^ at 1 cm^−1^ resolution, and 8 interferograms are added together before the Fourier transform. Spectra were collected from 4000 to 380 cm^−1^, and the absorbance band at approximately 1008 cm^−1^ was normalized based on 1 so that the absorbance was normalized between 0 and 1. The degrees of acetylation (DA) and deacetylation (DDA) of chitin and chitosan derived from mealworm larvae and shrimp shells were calculated using FT-IR absorbance. Using the ratio of absorbance at wavelengths of 1655 (A1655) and 3450 cm^−1^ (A3450) and the absorbance ratios of 1320 (A1320) and 1420 cm^−1^ (A1420), the degree of acetylation and deacetylation was calculated as follows [36].
DA = (A1655/A3450) × 115, DDA = A1320/A1420

### 3.8. Thermal Characterization of Chitin and Chitosan by Differential Scanning Calorimetry (DSC)

The thermal properties of chitin and chitosan were measured using DSC Q20 (TA Instruments, New Castle, DE, USA) [5]. The sample weighing 3 mg was placed in an aluminum pan (T-ZERO Hermetic Aluminum pan), which was sealed with a T-ZERO Hermetic upper die and T-ZERO Hermetic lower die. The flow rate of nitrogen was 50 mL/min, and the thermal properties of the samples were measured while continuously increasing the temperature from 25 °C to 400 °C at a rate of 10 °C/min. In addition, the exothermic peaks (*T_s_*, *T_max_*, *T_e_*) and endothermic peaks (*T_s_*, *T_max_*, *T_e_*) and the energy (Δ*H*) of each peak were measured.

### 3.9. Statistical Analysis

The experimental results were measured three times and expressed as mean ± standard deviation using SPSS Statistics (ver. 20.0, IBM Corp., IBM, Armonk, NY, USA). After analysis by ANOVA, the statistically significant differences between the experimental results were verified using Tukey’s multiple comparison test at a significance level of *p* < 0.05.

## 4. Conclusions

In this study, enzymes were used to extract eco-friendly and economical chitin from edible insects, and at this time, the RMS method was used as the optimum extraction condition. In addition, the physicochemical properties of the extracted chitin were investigated. As a result, the optimal conditions for protein removal hydrolysis were the substrate 7.5 g, enzyme 80 mL, and reaction time 24 h. The extracted mealworm chitin had similar physical properties to the extracted chitosan from the crustaceans sold as reagents. Moreover, the chitin extracted from the mealworm showed a higher yield and solubility than the control chitin. It is believed that it presented economic and eco-friendly chitin and chitosan extraction process guidelines and suggested the possibility as a new functional food raw material. From an industrial perspective, edible insect protein, a by-product, can be expected to be effective as a material for high-protein functional foods or cosmetics, such as functional materials that replace meat proteins, as well as the use of edible insect chitin as medi-food, such as that with anti-inflammatory, anti-obesity, and anti-cancer properties.

## Figures and Tables

**Figure 1 molecules-28-03254-f001:**
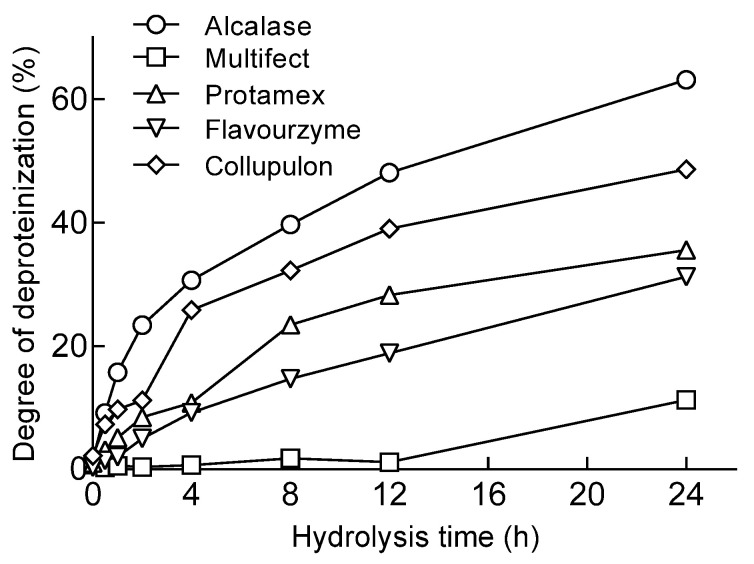
Changes in degree of deproteinization (DP) according to proteolytic enzymes during protein removal from mealworm larvae.

**Figure 2 molecules-28-03254-f002:**
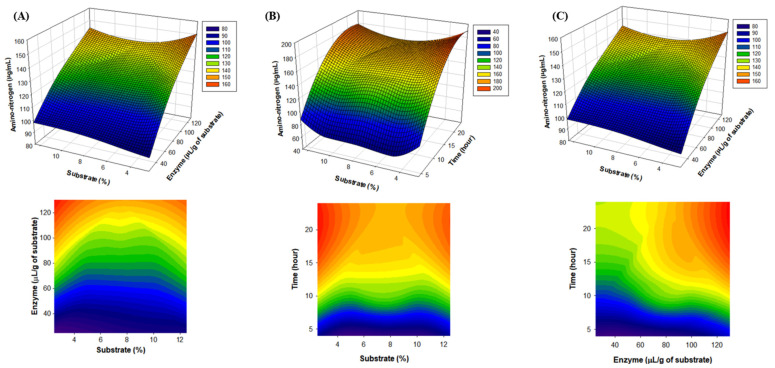
Response surface methodology (RSM)-run 3D graph. (**A**) Substrate (%), enzyme (μL/g of substrate), and amino-nitrogen (μg/mL) 3D graph. (**B**) enzyme (μL/g of substrate), time (hour), and amino-nitrogen (μg/mL) 3D graph. (**C**) Substrate (%), time (hour), and amino-nitrogen (μg/mL) 3D graph.

**Figure 3 molecules-28-03254-f003:**
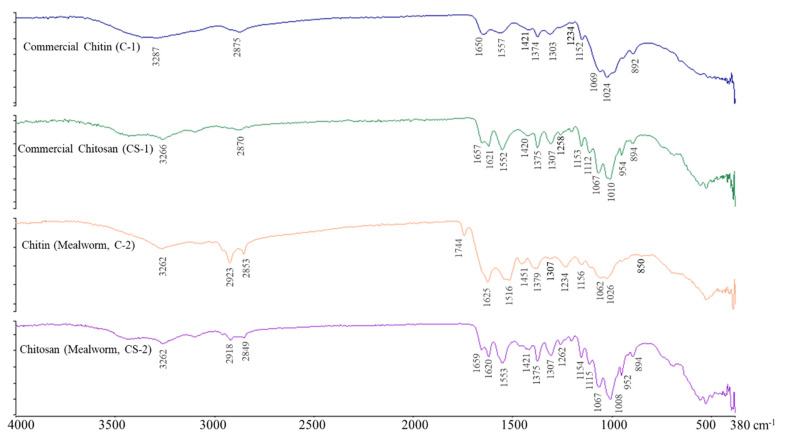
FT-IR spectrophotogram of commercial chitin (C-1), chitosan (CS-1), chitin (C-2), and chitosan (CS-2) from mealworm larvae.

**Table 1 molecules-28-03254-t001:** Comparison of RSM and actual results.

RSM Result				Actual Result			
Substrate (g)	Enzyme (μL)	Time (h)	Amino-Nitrogen (mg/g of Sample)	Substrate (g)	Enzyme (μL)	Time (h)	Amino-Nitrogen (mg/g of Sample)
7.5	80	24	163.17	7.5	80	24	162.99

**Table 2 molecules-28-03254-t002:** DSC of commercial chitin, mealworm chitin, commercial chitosan, and mealworm chitosan.

Sample	Exotherm (°C)				Endotherm (°C)			
	*T_s_*_1_ (°C)	*T_max_*_1_ (°C)	*T_e_*_1_ (°C)	Δ*H*_1_ (J/g)	*T_s_*_2_ (°C)	*T_max_*_2_ (°C)	*T_e_*_2_ (°C)	Δ*H*_2_ (J/g)
commercial chitin	140.77	143.29	157.37	−116.6	349.90	385.40	397.15	17.848
mealworm chitin	132.82	153.76	179.76	−91.08	325.22	359.82	394.48	24.075
commercial chitosan	140.77	143.65	165.68	−183.1	281.20	306.47	331.38	104.7
mealworm chitosan	141.13	147.63	197.81	−179.9	301.06	337.52	371.82	181.7

*T_s_* start temperature, *T_max_* maximum temperature, *T_e_* end temperature, Δ*H* thermal energy.

**Table 3 molecules-28-03254-t003:** Degree of deacetylation (%) and solubility (%) of chitosan extracted from mealworm when compared to commercial chitosan.

	Mealworm Chitosan	Commercial Chitosan
Degree of deacetylation (%)	37.34 ± 4.35	57.58 ± 7.88
Solubility (%)	45.94 ± 15.33	61.34 ± 15.96

Data are expressed as mean ± standard deviation (SD).

**Table 4 molecules-28-03254-t004:** Experimental design for chitin production.

Run Order	Coded Variables			Real Variables			Amino-Nitrogen (mg/g of Sample)
	X	Y	Z	X Substrate Concentration (%)	Y Enzyme Concentration (%)	Z Time (h)	
1	−1.00000	−1.00000	−1.00000	5	50	6	83.46 ± 0.33
2	0.00000	−1.68179	0.00000	7.5	25	8	94.93 ± 4.35
3	1.0000	−1.0000	−1.00000	10	50	6	83.49 ± 0.06
4	1.68179	0.00000	0.00000	12.5	80	8	122.03 ± 0.94
5	1.00000	1.00000	−1.00000	10	100	6	99.45 ± 3.48
6	0.00000	0.00000	1.68179	7.5	80	24	162.99 ± 4.68
7	0.00000	0.00000	0.00000	7.5	80	8	118.61 ± 0.63
8	0.00000	0.00000	0.00000	7.5	80	8	119.66 ± 1.60
9	1.00000	1.00000	1.00000	10	100	12	148.87 ± 3.34
10	0.00000	0.00000	0.00000	7.5	80	8	122.48 ± 1.77
11	−1.00000	−1.00000	−1.00000	5	50	6	78.42 ± 0.78
12	0.00000	0.00000	0.00000	7.5	80	8	121.15 ± 3.01
13	−1.68179	0.00000	0.00000	2.5	80	8	132.79 ± 1.38
14	0.00000	0.00000	0.00000	7.5	80	8	116.00 ± 2.08
15	−1.00000	1.00000	1.00000	5	100	12	157.99 ± 0.92
16	0.00000	0.00000	0.00000	7.5	80	8	120.07 ± 0.20
17	0.00000	0.00000	−1.68179	7.5	80	4	65.57 ± 1.74
18	0.00000	0.00000	0.00000	7.5	80	8	121.79 ± 3.02
19	−1.00000	−1.00000	1.00000	5	50	12	127.91 ± 1.14
20	1.00000	−1.00000	−1.00000	10	50	6	83.45 ± 5.49
21	0.00000	0.00000	0.00000	7.5	80	8	122.96 ± 2.51
22	0.00000	−1.68179	0.00000	7.5	25	8	97.14 ± 1.94
23	1.68179	0.00000	0.00000	12.5	80	8	130.52 ± 1.02
24	0.00000	0.00000	0.00000	7.5	80	8	129.01 ± 0.79
25	0.00000	0.00000	1.68179	7.5	80	24	162.01 ± 2.17
26	0.00000	0.00000	−1.68179	7.5	80	4	67.89 ± 3.08
27	0.00000	0.00000	0.00000	7.5	80	8	120.45 ± 1.74
28	−1.00000	1.00000	−1.00000	5	100	6	94.96 ± 0.41
29	1.00000	1.00000	−1.00000	10	100	6	99.25 ± 0.57
30	0.00000	1.68179	0.00000	7.5	130	8	136.45 ± 4.17
31	−1.00000	−1.00000	1.00000	5	50	12	130.30 ± 2.66
32	−1.68179	0.00000	0.00000	2.5	80	8	127.57 ± 1.31
33	1.00000	1.00000	1.00000	10	100	12	155.39 ± 1.99
34	0.00000	0.00000	0.00000	7.5	80	8	119.53 ± 2.70
35	0.00000	1.68179	0.00000	7.5	130	8	139.70 ± 2.11
36	1.00000	−1.00000	1.00000	10	50	12	132.45 ± 4.34
37	1.00000	−1.00000	1.00000	10	50	12	131.45 ± 2.71
38	−1.00000	1.00000	−1.00000	5	100	6	94.92 ± 2.38
39	0.00000	0.00000	0.00000	7.5	80	8	125.42 ± 1.43
40	−1.00000	1.00000	1.00000	5	100	12	159.73 ± 1.60

## Data Availability

The data that support the findings of this study are available from the corresponding author upon reasonable request.

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
