# Peer review of "The Preparation and Physiochemical Characterization of Tenebrio molitor Chitin Using Alcalase"

_molecules, 2023, doi:10.3390/molecules28073254_

Round 1

Reviewer 1 Report

The manuscript entitled "The Preparation and Physiochemical Characterization of Tenebrio molitor Chitin using Alcalase" is a quite interesting piece of scientific work. What must be highlighted, the work in general is well designed, especially the RSM approach, whose description and analysis of results are comprehensive.

However, I have a few issues with the manuscript, which the authors should address:

1. The introduction is too short and narrow. In order to attract readers, it should be expanded. Why is it necessary to use chitin, and why should new acquisition methods be designed or discovered? Moreover, there is nothing said about the actual drawbacks of using the chemical methods of chitin extraction. Authors should put greater emphasis on why their method is an attractive alternative to conventional processes.

2. Numeration of subsections and tables Please check carefully for the correct numeration in your manuscript. There are two different tables with no. 4. Moreover, materials and methods are annotated with no. 3, but the whole chapter is numbered with 4.x (except for the 3.1 Materials).

3. There is no Conclusions section in your manuscript. That is unacceptable. Please write a comprehensive piece of text.

Author Response

* The changes in the revised manuscripts are shown in red. This manuscript is a file that can track the modified ones.

Dear Reviewer

Thank you very much for your valuable comments and suggestions. We have incorporated all the suggestions made by the review and the list of itemized revision is attached herewith. We would appreciate it very much if you take good consideration of our revision for publication in Molecules.

Yours sincerely  

Reviewer 2 Report

The manuscript study "The Preparation and Physiochemical Characterization of Te- 2

nebrio molitor Chitin using Alcalase". The idea and information provided are interesting. Below are some of the technical and non-technical points which should be addressed in order to move on:

In this study, hydrolysis was done with different enzymes. Why did you only mention alcalase in the title?

Abstract: The authors should reorganize as following order: (Problem of research, aim of study, remarkable methodology, remarkable results, and significance of study).

Why were these enzymes chosen? purpose?

The goals and importance of your work are not clear. What are its practical results?

It is better to provide more data in the abstract (key results).

How were the optimal conditions obtained? It is better to specify these items in the abstract.

It is suggested to express the potential of industrial applications of this study.

The necessity and innovation of this study are not clear!

Conclusions: The authors should write this section in comprehensive and suitable style.

Mention the key results of other similar research.

Compare the results of this study with other findings.

Mention the scientific reasons for the findings in more detail.

Author Response

(The authors gave the same response as above.)

Reviewer 3 Report

·         In abstract, authors compare the numerical solubility and DDA obtained with commercial types. What rationale/conclusion can be made from there?

·         Review the chitin from other insects such as black soldier fly larvae in introduction. Also, what is the disadvantages of chitin from crustacean that led to the current study?

·         From the 3D plot RSM, the red regions seem limited. It is because of the ranges selected were deviated?? Please justify further.

·         Also, the 3D plots are not clear. Can you replace it?

·         Can you show the statistical model obtained from RSM?

·         Also discuss the ANOVA from RSM and show the lack of fit.

·         Include thermodynamic results in Abstract.

·         Materials and Methods should come first before discussion.

·         How did you determine the factors for RSM ? elaborate.

·         Also, how did you determine the ranges for the factors?? elaborate.

·         You can remove eq from section 4.4. It is not helpful and general model only.

·         Overall the works are fine can be considered for publications after addressing the comments.

Author Response

(The authors gave the same response as above.)

Round 2

Reviewer 1 Report

The manuscript can be accepted in its current form.